# The Correlation amongst Salty Taste Preference and Overactive Bladder Symptoms in Female Individuals

**DOI:** 10.3390/ijerph18020518

**Published:** 2021-01-10

**Authors:** Jin-Won Noh, Kyoung-Beom Kim, Jae Heon Kim, Young Dae Kwon

**Affiliations:** 1Department of Health Administration, Dankook University, Cheonan 31116, Korea; jinwon.noh@gmail.com (J.-W.N.); aefile01287@korea.ac.kr (K.-B.K.); 2Department of Urology, Soonchunhyang University Seoul Hospital, Soonchuhyang University Medical College, 59, Daesagwan-ro, Yongsan-gu, Seoul 140-743, Korea; 3Department of Humanities and Social Medicine, College of Medicine and Catholic Institute for Healthcare Management, The Catholic University of Korea, Seoul 06591, Korea; healthcare@catholic.ac.kr

**Keywords:** sodium, dietary, overactive bladder, lower urinary tract symptoms

## Abstract

Sodium intake could have an indirect effect on storage symptoms as it is an established fact that it has a correlation to hypertension (HTN). However, clinical support for the correlation of sodium intake to STORAGE symptom remains scarce. Therefore, the present work sought to determine how sodium intake and OAB symptom seriousness were correlated in female individuals based on a cross-sectional research design. Data from 115,578 respondents chosen from 228,921 individuals enrolled in the Korean Community Health Survey (KCHS) were subjected to cross-sectional analysis. The correlation amongst sodium intake and STORAGE symptom was assessed on the basis of estimates of the incidence rate ratio (IRR) with 95% confidence interval (CI). Furthermore, joint correspondence analysis (JCA) was conducted to investigate how a predilection for salty taste was associated with several factors, including STORAGE symptom, socio-economic factors, comorbidities, and dietary habits. By comparison to respondents indicating a neutral predilection for salty taste, those who indicated a predilection for bland and salty taste were 7.1% (IRR = 1.071, *p* < 0.001, 95% CI = 1.045–1.097) and 20.5% (IRR = 1.205, *p* < 0.001, 95% CI = 1.177–1.234) more probable to experience severe STORAGE symptom, within an adjusted model with multiple variables. JCA confirmed the existence of a correlation between predilection for salty taste and STORAGE symptom. A close correlation was established to exist between predilection for salty taste and lower urinary tract symptoms (LUTS), such as voiding and storage symptoms and nocturia. Moreover, sodium intake was found to be a risk factor for LUTS seriousness in both excessive and deficient amount.

## 1. Introduction

Both nutrients of fruit and vegetable origin and micronutrients serve as antioxidants and influence lower urinary tract symptoms (LUTS) [1,2]. The topic has been investigated in a few studies, but the findings have been inconsistent, particularly in relation to LUTS. The latest studies in the field of medicine have extensively explored sodium intake in relation to various conditions, such as hypertension (HTN), cardiovascular disease (CVD), and chronic kidney disease (CKD) [3,4]. Due to its direct association with CVD-related death, sodium intake is a significant health matter worldwide [5,6].

There are two aspects that can clarify the potential correlation amongst sodium intake and LUTS, namely, the indirect impact of sodium intake through its influence on HTN [4,7] and the direct impact on the epithelial sodium channel of the bladder [8,9]. Sodium intake is such an important risk factor for HTN occurrence or worsening that it has led to the classification of individuals with HTN into two groups with and without sensitivity to sodium intake. 

In cases of LUTS, those with HTN showed a greater preponderance of urinary storage symptom compared to those who did not suffer from HTN [3]. Sodium intake acts indirectly in the context of LUTS, causing the autonomic nerve system to become hyperactive, particularly with regard to prostate and bladder innervation [10]. Furthermore, alpha blockers may become less effective therapeutically due to HTN triggered by sodium intake [8]. It was found that, of the various nutrients, protein intake and sodium intake respectively constituted a risk factor for worsening voiding symptom and storage symptom [2,11]. Meanwhile, empirical work is the basis for revealing the direct impact of sodium intake within a LUTS context [4]. Storage symptoms can be stimulated by elevated sodium intake through epithelial sodium channel upregulation.

A possible correlation between sodium intake and LUTS, particularly overactive bladder (OAB) and other storage symptoms, has been signalled by earlier our research [12]. Moreover, evidence has been produced regarding a close association between a predilection for salty taste and LUTS, such as voiding and storage symptoms and nocturia, with sodium being a risk factor for LUTS seriousness in both excessive and deficient amount [12]. However, this correlation has so far not been investigated in the case of female subjects. 

The present paper explores the hypothesis that sodium predilection may have an indirect effect on storage symptom by causing disruptions to the circulation system, such as excessive activation of the adrenergic nerve system and direct bladder epithelium stimulation. To this end, analysis focuses on how sodium predilection and storage symptom are correlated after consideration of various socioeconomic factors. 

## 2. Methods

### 2.1. Data and Subjects

This study utilized data obtained from the 2012 Korean Community Health Survey (KCHS). The KCHS was conducted annually since 2008 organized by the Korean Centers for Disease Control and Prevention. The objectives of KCHS to produce community-based comparable health statistics for evaluation of diseases prevention program and community health promotion. To extract representative samples, the KCHS used a multistage sampling design [13]. This study was carried out using data from 115,578 participants, after excluding 113,343 respondents those who male (*n* = 102,898), and who have missing data in Overactive Bladder Symptom Score (OABSS), salty taste preferences, or covariates (*n* = 10,445) from the 228,921 participants in the 2012 KCHS (Figure 1).

### 2.2. Measurements

The outcome variable, storage symptoms based on responses from the Korean version of OABSS Questionnaire on KCHS [14]. The OABSS is comprised of four questions that assess symptoms of storage, daytime frequency (How many times do you typically urinate from waking in the morning until sleeping at night?), night time frequency (How many times do you typically wake up to urinate from sleeping at night until waking in the morning?), urgency (How often do you have a sudden desire to urinate, which is difficult to defer?), and urge incontinence (How often do you leak urine because you cannot defer the sudden desire to urinate?) The total score ranges from 0 to 15, higher score indicates severe storage symptoms. The total score was categorized into mild (≤5), moderate (6 to 11), or severe (12≤) storage symptoms [15]. The exposure variable, salty taste preference was measured by self-rated 5-point Likert scale. It was categorized into (very) salty, neutral, and (very) blandly.

Covariates considered socio-demographic factors, comorbidities, and dietary behaviors. Socio-demographics included age, marital status, education level, household income, and residential area. Age was consisting of seven groups from “19 to 29 years old” in subsequent 10-year intervals up to “90 years old or older.” Marital status was classified into three categories as corresponding to “married”, “separated, divorced, or widowed”, or “never married.” Education level were based on highest level completed and grouped as “elementary school graduate or lower”, “middle school graduate”, “high school graduate”, and “college graduate or higher.” Household income was divided into quartile. Residential area was classified into “capital”, “urban”, and “rural”, based on 16 governmental administrative districts. In consideration of comorbidities, hypertension, diabetes, and dyslipidemia which known as highest degree of prevalence rate among adults were included [16]. Information of comorbidities was ascertained through self-report of a physician’s diagnosis. Dietary behaviors included breakfast eating. it was ascertained using the question, “How many days do you eat the breakfast in last week?” Response was categorized as “5–7 days”, “1–4 days”, or “Never eat in last week.”

### 2.3. Statistical Analysis

To summarize the socio-demographic, comorbidities, and dietary behavior characteristics of study participants according to salty taste preference, descriptive analysis was performed. The frequency and weighted proportion were reported as descriptive statistics. A Chi-squared test was performed to identify the proportion differences by salty taste preference among OABSS and covariates.

To explore the relationship between salty taste preference, storage symptom, socioeconomic, comorbidity, and dietary behavior in multifactorial manner, a joint correspondence analysis (JCA) was carried out. The JCA visualizes degree of clustering of points in coordinate plot at the level of the response categories of each variable [17]. As guidelines for interpreting the relationships between variables, degree of clustering of points on the JCA plot in terms of their angle from the centroid and points in the same quadrant can be used [18]. To investigate the relationship between OABSS and salty taste preference among Korean female adults, two sets of negative binomial regression models were fitted. The model I adjusted for age, and model II additionally adjusted for marital status, education level, household income, residential area, hypertension, diabetes mellitus, dyslipidemia, and breakfast eating. The adjusted incidence rate ratio (IRR) with 95% confidence intervals (CI) were reported. A sample design and benchmark weight were employed in analyses to account for complex sampling design of KCHS and ensure the produced estimates are nationally representative of the South Korean female population. The data were analyzed using Stata/MP 16.1 (StataCorp LP, College Station, TX, USA) and α = 0.05 (two-tailed) were considered to determine the statistical significance.

### 2.4. Ethics Statement

Procedures of this study were reviewed and approved by the Institutional Review Board of Soonchunhyang University Seoul Hospital with a waiver for informed consent (2018-12-011). The study protocol of KCHS was reviewed and approved by the institutional review board of Korean Centers for Disease Control and Prevention (2010-02CON-22-P). The privacy risk to the participants is minimized because KCHS data was anonymized. The KCHS data is openly accessible at national public repository (http://chs.cdc.go.kr). 

## 3. Results

Table 1 summarizes the general characteristics of study participants by salty taste preference. Among 115,578 South Korean female study participants, 60,533 (51.8%) were reported “neutral”, 27,292 (24.4%) were “blandly”, and 27,753 (23.8%) were “salty” as their salty taste preference. About 6809 (5.9%) of participants had moderate/severe STORAGE symptoms. A relatively small proportion of participants reported their salty taste preference as “blandly” in those who younger (age 19–29) or older (90 or higher) compared with middle-aged, unmarried, low-educated, in poverty, live in urbanized area, having comorbidities, skipping breakfast, having more frequent storage symptoms, and severe storage severity. The distribution of salty taste preference was significantly differed by age, marital Status, education Level, household income, residential area, hypertension, diabetes, dyslipidemia, frequency of weekly breakfast eating, storage symptoms (daytime frequency, nighttime frequency, urgency, and urge incontinence), and storage severity (*p* < 0.05).

A JCA coordinate plot illustrates multifactorial relationship between overactive bladder symptom, salty taste preference, socioeconomic, comorbidity, and dietary behavior. The left-lower quadrant shows the association between respondents those who prefer salty taste and moderate or severe STORAGE symptom. On the other side, participants who reported their salty taste preference as blandly or neutral, and mild storage symptom were clustered in the right-upper quadrant, close to the centroid. It was identified that salty taste preference was associated with storage symptom (Figure 2).

Result from multivariable adjusted negative binomial model to investigate the relationship between OABSS and salty taste preference among Korean female adults suggests that participants those who prefer blandly, or salty taste preference were both associated with increased OABSS. Participants who reported their salty taste preference as blandly and salty were indicates 7.1% (IRR = 1.071, *p* < 0.001, 95% CI = 1.045 to 1.097) and 20.5% (IRR = 1.205, *p* < 0.001, 95% CI = 1.177 to 1.234) increased likelihood of worse OABSS condition compared with participants those who reported their salty taste preference as neutral in multivariable adjusted model (Table 2). 

## 4. Discussion

It is an established fact that sodium predilection worsens HTN and makes it more likely for CVD patients to die, so it can affect the circulatory system and therefore demands close attention [19,20]. HTN, metabolic syndrome, atherosclerosis, fatty liver, and storage symptom are all interconnected. Hence, it can be said that storage symptom is subject to the influence of both HTN and other elements of the circulatory system.

The evidentiary support that has been generated so far about the correlation amongst sodium intake/predilection and LUTS seriousness is not extensive and stems primarily from research on male individuals. Therefore, the present work seeks to expand the topic to female individuals. In cross-sectional examination of a population sampled arbitrarily revealed that sodium intake and LUTS were significantly positively correlated 2. A trend that was particularly significant for storage LUTS was that the probability of graver LUTS has higher in male individuals consuming elevated amount of sodium (OR = 2.25; 95% CI = 1.26–4.03), but no definitive correlation was established regarding voiding LUTS. In a different study with case-control design, BPH open to surgical treatment was more likely to be diagnosed (OR = 1.30) in cases of sodium intake [11].

The International Prostate Symptom Score (IPSS) applied in our previous work confirmed that sodium intake and storage symptom were correlated in male [12]. More specifically, unlike the group with neutral sodium predilection, the group with a predilection for salty taste demonstrated a significant correlation not only with elevated IPSS total score (Coef = 0.31; 95% CI: 0.27, 0.35), but also heightened risk of serious IPSS grade (OR = 1.46; 95% CI:1.35, 1.57), heightened IPSS voiding score (Coef = 0.38; 95% CI: 0.32, 0.44), heightened IPSS storage score (Coef = 0.25; 95% CI: 0.22, 0.29), as well as being more likely to develop symptoms of IPSS nocturia (OR = 1.21; 95% CI:1.16, 1.27).

In a recent comprehensive cross-sectional study, a positive correlation was established between sodium intake per day and both day-time and night-time frequency. This correlation was attributed primarily to polydipsia associated with sodium intake as this elevated blood osmotic pressure. Furthermore, the study did not undertake sex-based subgroup analysis, even though, of the total of 321 participants employed, 219 were female [21].

In the present study, by comparison to respondents indicating a neutral predilection for salty taste, those who indicated a predilection for bland and salty taste were 7.1% (IRR = 1.071, *p* < 0.001, 95% CI = 1.045–1.097) and 20.5% (IRR = 1.205, *p* < 0.001, 95% CI = 1.177–1.234) more probable to experience severe storage symptom, within an adjusted model with multiple variables. Furthermore, JCA was conducted as multifactorial analysis, confirming the existence of a correlation between predilection for salty taste and both moderately and highly severe storage symptom.

Supported by a number of empirical studies [4,9], sodium intake stimulating the bladder epithelium directly is the traditional interpretation given for the correlation between sodium intake and storage symptom. According to Yamamoto and colleagues, the bladder epithelium sodium channel is upregulated as a result of elevated sodium intake [4]. Moreover, storage symptom worsening when bladder afferent pathways are irregularly activated can be attributed to the discharge of neurotransmitters and other bioactive substances from the bladder epithelium when the latter is stimulated [22]. It is noteworthy that the upregulation of the sodium channel of the bladder epithelium and urinary storage symptom are closely correlated [23].

Despite its significant results, the present study is not without shortcomings. One major shortcoming is that the causal effect of sodium intake on OAB seriousness cannot be determined through cross-sectional analysis. Another shortcoming is that the work did not include BMI data and, so far, BMI and OAB seriousness have not been definitively correlated, despite some evidentiary support for the association between obesity and LUTS seriousness [24]. Additionally, the use of a subjective questionnaire to establish the extent of sodium intake constitutes a shortcoming as well. Previous research emphasised the significance of determining sodium levels in urine for accurately quantifying the sodium intake extent [6]. The levels of urinary sodium were not measured in the present work, but the correlation amongst these levels and self-appraised sodium predilection scale has been proven in other studies. For instance, a self-evaluation questionnaire comprising 127 items about dish frequency indicated that sodium predilection, real sodium intake, and likelihood of increased sodium intake were significantly correlated [25]. Similarly, a correlation was found between salty taste thresholds in healthy individuals and individuals with non-dialysis CKD and salty taste thresholds and sodium levels in urine [26]. Moreover, our study is providing lacking information about OAB symptoms secondary to bladder outlet obstruction due to pelvic organ prolapse that is frequently observed in obese and old women [27].

A final shortcoming is that the study did not include participants with inadequate dietary customs owing to lack of data about socio-economic factors. Nutritional quality or dietary habits are likely to be suboptimal in individuals who do not disclose income or education information as they usually have a low socio-economic standing.

## 5. Conclusions

A correlation was established between sodium preference and OAB seriousness, with both excessive and poor sodium intake contributing to LUTS worsening by comparison to neutral sodium intake. This study showed indirect evidence of relationship between sodium intake and OAB severity. Furthermore, sodium preference was found to have an association with lifestyle and socio-economic factors as well. To solidify these findings, further prospective studies must be conducted.

## Figures and Tables

**Figure 1 ijerph-18-00518-f001:**
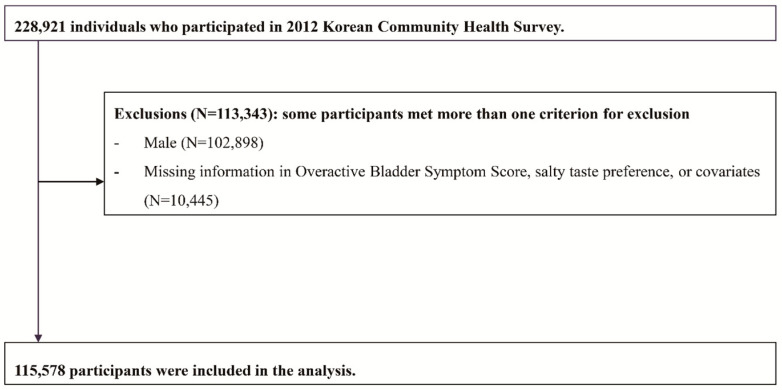
Inclusion and exclusion in the study population.

**Figure 2 ijerph-18-00518-f002:**
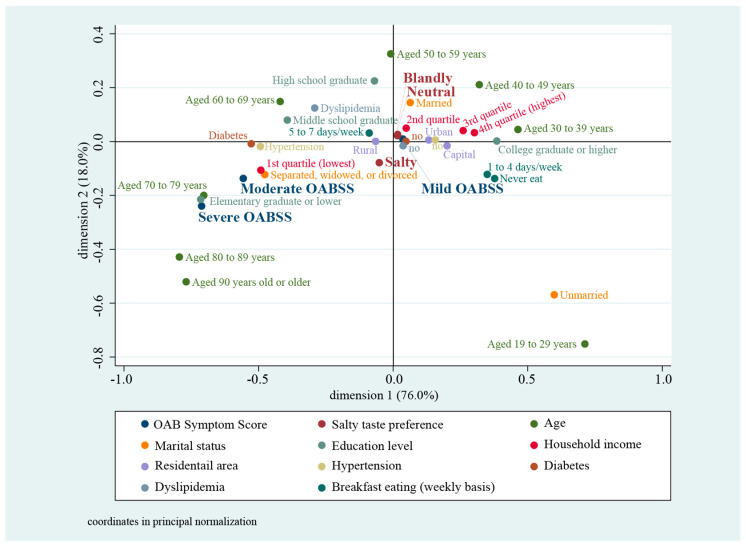
Multifactorial relationship between overactive bladder symptom, salty taste preference, socioeconomic, comorbidity, and dietary behavior. OABSS, overactive bladder symptom score.

**Table 1 ijerph-18-00518-t001:** Characteristics of study participants by salty taste preference.

Variable	Subcategory	Salty Taste Preference						Total	Test Statistic	*p*-Value
		Neutral		Blandly		Salty				
		(*n* = 60,533)		(*n* = 27,292)		(*n* = 27,753)				
		*n*	Proportion	*n*	Proportion	*n*	Proportion	*n*		
Age	19–29	6017	47.6	2618	20.7	4002	31.7	12,637	73.361	<0.001
	30–39	9511	53	4625	25.8	3815	21.3	17,951		
	40–49	12,206	56.3	5307	24.5	4180	19.3	21,693		
	50–59	12,443	55.3	5647	25.1	4392	19.5	22,482		
	60–69	9465	51.9	3945	21.6	4833	26.5	18,243		
	70–79	8101	48.6	3695	22.2	4875	29.2	16,671		
	80–89	2555	47.4	1325	24.6	1515	28.1	5395		
	90 or higher	235	46.4	130	25.7	141	27.9	506		
Marital Status	Married	40,945	54.3	18,119	24	16,318	21.6	75,382	151.429	<0.001
	Separated, divorced, widowed	13,138	49	6330	23.6	7359	27.4	26,827		
	Unmarried	6450	48.2	2843	21.3	4076	30.5	13,369		
Education Level	Elementary graduate or lower	10,072	47.9	4370	20.8	6589	31.3	21,031	75.288	<0.001
	Middle school graduate	10,716	53	4170	20.6	5326	26.4	20,212		
	High school graduate	6865	55.4	2816	22.7	2709	21.9	12,390		
	College graduate or higher	32,880	53.1	15,936	25.7	13,129	21.2	61,945		
Household income	1st Quartile (lowest)	16,326	50.2	7263	22.3	8944	27.5	32,533	22.796	<0.001
	2nd Quartile	16,770	53.1	7343	23.3	7452	23.6	31,565		
	3rd Quartile	13,920	54	6107	23.7	5731	22.2	25,758		
	4th Quartile (highest)	13,517	52.6	6579	25.6	5626	21.9	25,722		
Residential area	Capital	5882	51.8	2887	25.4	2597	22.8	11,366	2.617	0.035
	Urban	11,789	51.6	5606	24.5	5450	23.9	22,845		
	Rural	42,862	52.7	18,799	23.1	19,706	24.2	81,367		
Hypertension	No	46,748	53.3	20,789	23.7	20,155	23	87,692	37.078	<0.001
	Yes	13,785	49.4	6503	23.3	7598	27.2	27,886		
Diabetes	No	55,878	52.7	24,833	23.4	25,318	23.9	106,029	11.9	<0.001
	Yes	4655	48.7	2459	25.8	2435	25.5	9549		
Dyslipidemia	No	54,013	52.6	24,233	23.6	24,396	23.8	102,642	4.452	0.012
	Yes	6520	50.4	3059	23.6	3357	26	12,936		
Breakfast eating (weekly)	5–7 days	48,928	52.8	22,685	24.5	21,140	22.8	92,753	122.33	<0.001
	1–4 days	6813	52.1	2618	20	3642	27.9	13,073		
	Never	4792	49.1	1989	20.4	2971	30.5	9752		
OABSS—daytime frequency	≤7	47,237	53.3	20,644	23.3	20,700	23.4	88,581	20.352	<0.001
	8~14	12,589	49.7	6223	24.6	6534	25.8	25,346		
	≥15	707	42.8	425	25.7	519	31.4	1651		
OABSS—nighttime frequency)	0	26,636	53.8	12,218	24.7	10,672	21.5	49,526	33.066	<0.001
	1	20,560	53.4	8856	23	9099	23.6	38,515		
	2	8227	49.8	3753	22.7	4536	27.5	16,516		
	≥3	5110	46.4	2465	22.4	3446	31.3	11,021		
OABSS—urgency	None	51,924	53.2	23,187	23.8	22,403	23	97,514	24.416	<0.001
	Less than once a week	4702	49.8	2081	22	2655	28.1	9438		
	Once a week or more	1895	46.8	910	22.5	1244	30.7	4049		
	About once a day	955	46.2	486	23.5	626	30.3	2067		
	2–4 times a day	687	41.5	403	24.4	564	34.1	1654		
	5 times a day or more	370	43.2	225	26.3	261	30.5	856		
OABSS—urge incontinence	None	55,147	53	24,653	23.7	24,319	23.4	104,119	15.789	<0.001
	Less than once a week	3052	49	1402	22.5	1772	28.5	6226		
	Once a week or more	1186	46.3	577	22.5	801	31.2	2564		
	About once a day	575	43.7	305	23.2	435	33.1	1315		
	2–4 times a day	400	41.5	258	26.8	305	31.7	963		
	5 times a day or more	173	44.2	97	24.8	121	30.9	391		
OAB severity	Mild	57,529	52.9	25,655	23.6	25,585	23.5	108,769	33.409	<0.001
	Moderate	2645	44.5	1412	23.8	1883	31.7	5940		
	Severe	359	41.3	225	25.9	285	32.8	869		

OAB, overactive bladder; OABSS, overactive bladder symptom score.

**Table 2 ijerph-18-00518-t002:** Impact of salty taste preference on overactive bladder symptom among South Korean female adults.

Salty Taste Preference	Overactive Bladder Symptom Score (Score Range = 0 to 15)									
	Age-Adjusted					Multivariable Adjusted				
	IRR	Linearized SE	*p*-Value	95% CI		IRR	Linearized	*p*-Value	95% CI	
							SE			
Neutral	(Ref)	(Ref)								
Blandly	1.065	0.013	<0.001	1.039	1.092	1.071	0.013	<0.001	1.045	1.097
Salty	1.236	0.012	<0.001	1.207	1.267	1.205	0.015	<0.001	1.177	1.234

IRR, incidence rate ratios; CI, confidence interval; SE, standard error; Ref, reference. The KCHS as a sample survey was analyzed by study subject and with applied weight calculated in production of the sample design weight and benchmark weight. Strata with single sampling unit centered at overall mean. Sample size = 115,578; weighted = 18,388,103. Multivariable model was additionally adjusted for marital status, education level, household income, residential area, hypertension, diabetes mellitus, dyslipidemia, and breakfast eating.

## Data Availability

Not applicable.

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
