# Peer review of "The Correlation amongst Salty Taste Preference and Overactive Bladder Symptoms in Female Individuals"

_ijerph, 2021, doi:10.3390/ijerph18020518_

Round 1

Reviewer 1 Report

  • Authors stated that “causal effect of sodium intake on OAB seriousness cannot be determined through cross-sectional analysis”. The study did not calculate the salt intake or the daily urinary sodium. The paper analyzed only the salty preference by a questionnaire, therefore the title of the paper is misleading and must be reconsidered. I suggest “ The correlation amongst salty taste preference and overactive bladder symptoms in female individuals”
  • Introduction section: must be edited. In particular the first 4 sentences are not  clear and not logically connected.
  • Authors should edited: overactive bladder (OAB) symptoms. [as plural and not singular] and not symptom (referring to storage symptoms)
  • I suggest to add: Another limitation of the study is the lacking information about OAB symptoms secondary to bladder outlet obstruction due to pelvic organ prolapse that is frequently observed in obese and old women.
  • Ref

    de Boer TA, Salvatore S, Cardozo L, Chapple C, Kelleher C, van Kerrebroeck P, Kirby MG, Koelbl H, Espuna-Pons M, Milsom I, Tubaro A, Wagg A, Vierhout ME. Pelvic organ prolapse and overactive bladder. Neurourol Urodyn. 2010;29(1):30-9. 

Author Response

  • Authors stated that “causal effect of sodium intake on OAB seriousness cannot be determined through cross-sectional analysis”. The study did not calculate the salt intake or the daily urinary sodium. The paper analyzed only the salty preference by a questionnaire, therefore the title of the paper is misleading and must be reconsidered. I suggest “ The correlation amongst salty taste preference and overactive bladder symptoms in female individuals”

: Thank you for your scrupulous review. According to your suggestion, we changed our title and as follows: The correlation amongst salty taste preference and overactive bladder symptoms in female individuals.

  • Introduction section: must be edited. In particular the first 4 sentences are not clear and not logically connected.

: Thank you for your kind comment. According to your suggestion, we changed the first 4 sentences.

  • Authors should edited: overactive bladder (OAB) symptoms. [as plural and not singular] and not symptom (referring to storage symptoms)

: Thank you for precise review. According to your suggestion, we changed the relevant terminology as storage symptoms in whole manuscript.  

  • I suggest to add: Another limitation of the study is the lacking information about OAB symptoms secondary to bladder outlet obstruction due to pelvic organ prolapse that is frequently observed in obese and old women.

: Thank you for your kind comment. According to your suggestion, we added those contents in limitation section and also we added the relevant reference (de Boer TA, Salvatore S, Cardozo L, Chapple C, Kelleher C, van Kerrebroeck P, Kirby MG, Koelbl H, Espuna-Pons M, Milsom I, Tubaro A, Wagg A, Vierhout ME. Pelvic organ prolapse and overactive bladder. Neurourol Urodyn. 2010;29(1):30-9.) and as follows: Moreover, our study is providing lacking information about OAB symptoms secondary to bladder outlet obstruction due to pelvic organ prolapse that is frequently observed in obese and old women [27].

Reviewer 2 Report

First of all I would like to appreciate the possibility to review this interesting paper concerning the possible link between sodium intake and OAB symptoms in female individuals, The number of analyzed patients included into this study is really impressive and in fact gives to the riders variable information’s concerning real OAB occurrence in South Korea population in various age groups.

 In the title and in Abstract Authors clearly stated that subjects of their study are female individuals but how it corresponds to Keywords: sodium; dietary; prostatic hyperplasia; lower urinary tract symptoms.  This might suggest that females have prostatic  hyperplasia???? Which of course is not the case. Definitely prostatic hyperplasia should be removed from Keywords!!!

Line  38 - and influence how prostate cells grow and differentiate. This statement at the beginning of an article concerning LUTS in females is also unfortunate and should be moved to discussion section

Line 74 - To this end, analysis focuses on how sodium predilection and OAB symptom are correlated as well as on the potential mediating effect of fruit and vegetable consumption. I cannot find in the article any data concerning fruit and vegetable consumption by study subjects as it was stated by Authors – please explain!!!

Line 114 Dietary behaviors included breakfast eating. it was ascertained using the question, “How many days do you eat the breakfast in last week?” Response was categorized as “5–7 days”, “1–4 days”, or “Never eat in last week.” It seems to me that only this question without information’s concerning lunch and dinner cannot give the Authors the real data about patients dietary behaviors. On the other hand it will be interesting to explain why breakfast eating is so important in this analysis especially in terms of LUTS occurrence? Please explain!

Line 226 Conclusions. A correlation was established between sodium intake and OAB seriousness, with both excessive and poor sodium intake contributing to LUTS worsening by comparison to neutral sodium intake. Furthermore, sodium intake was found to have an association with lifestyle and socio-economic factors as well. To solidify these findings, further prospective studies must be conducted.

I cannot agree with such conclusions. Authors did not measured sodium intake and their conclusions are based on predilection to salty taste which is only indirect measurement of salt intake and therefore conclusions should be modified.

Author Response

First of all I would like to appreciate the possibility to review this interesting paper concerning the possible link between sodium intake and OAB symptoms in female individuals, The number of analyzed patients included into this study is really impressive and in fact gives to the riders variable information’s concerning real OAB occurrence in South Korea population in various age groups.

 In the title and in Abstract Authors clearly stated that subjects of their study are female individuals but how it corresponds to Keywords: sodium; dietary; prostatic hyperplasia; lower urinary tract symptoms.  This might suggest that females have prostatic  hyperplasia???? Which of course is not the case. Definitely prostatic hyperplasia should be removed from Keywords!!!

 : Thank you for your kind comment. According to your suggestion, we changed the keyword.

Line  38 - and influence how prostate cells grow and differentiate. This statement at the beginning of an article concerning LUTS in females is also unfortunate and should be moved to discussion section

: Thank you for your kind comment. According to your suggestion, we removed the sentence.

Line 74 - To this end, analysis focuses on how sodium predilection and OAB symptom are correlated as well as on the potential mediating effect of fruit and vegetable consumption. I cannot find in the article any data concerning fruit and vegetable consumption by study subjects as it was stated by Authors – please explain!!!

: Thank you for your kind and precise comment. According to your suggestion, we removed the relevant sentence and change as follows: To this end, analysis focuses on how sodium predilection and storage symptom are correlated after consideration of various socioeconomic factors.

Line 114 Dietary behaviors included breakfast eating. it was ascertained using the question, “How many days do you eat the breakfast in last week?” Response was categorized as “5–7 days”, “1–4 days”, or “Never eat in last week.” It seems to me that only this question without information’s concerning lunch and dinner cannot give the Authors the real data about patients dietary behaviors. On the other hand it will be interesting to explain why breakfast eating is so important in this analysis especially in terms of LUTS occurrence? Please explain!

 : Thank you for your kind and precise comment. We used KCHS data which used common questionnaire form to check socioeconomic and lifestyle factors which includes only patter of breakfast.

Line 226 Conclusions. A correlation was established between sodium intake and OAB seriousness, with both excessive and poor sodium intake contributing to LUTS worsening by comparison to neutral sodium intake. Furthermore, sodium intake was found to have an association with lifestyle and socio-economic factors as well. To solidify these findings, further prospective studies must be conducted.

I cannot agree with such conclusions. Authors did not measured sodium intake and their conclusions are based on predilection to salty taste which is only indirect measurement of salt intake and therefore conclusions should be modified.

: Thank you for your kind and precise comment. We totally agree with you and according to your suggestion, we changed the conclusion and as follows: A correlation was established between sodium preference and OAB seriousness, with both excessive and poor sodium intake contributing to LUTS worsening by comparison to neutral sodium intake. This study showed indirect evidence of relationship between sodium intake and OAB severity. Furthermore, sodium preference was found to have an association with lifestyle and socio-economic factors as well. To solidify these findings, further prospective studies must be conducted.

Round 2

Reviewer 2 Report

No further cimments